# Multiple Myeloma: An Overview of the Current and Novel Therapeutic Approaches in 2020

**DOI:** 10.3390/cancers12102885

**Published:** 2020-10-08

**Authors:** Arthur Bobin, Evelyne Liuu, Niels Moya, Cécile Gruchet, Florence Sabirou, Anthony Lévy, Hélène Gardeney, Laly Nsiala, Laura Cailly, Stéphanie Guidez, Cécile Tomowiak, Thomas Systchenko, Vincent Javaugue, Géraldine Durand, Xavier Leleu, Mathieu Puyade

**Affiliations:** 1Department of Hematology, and CIC1402 INSERM Unit, Poitiers University Hospital, 2 Rue de la Milétrie, 86021 Poitiers, France; arthur.bobin@chu-poitiers.fr (A.B.); niels.moya@chu-poitiers.fr (N.M.); cecile.gruchet@chu-poitiers.fr (C.G.); florence.sabirou@chu-poitiers.fr (F.S.); anthony.levy@chu-poitiers.fr (A.L.); helene.gardeney@chu-poitiers.fr (H.G.); laly.nsiala@chu-poitiers.fr (L.N.); laura.cailly@chu-poitiers.fr (L.C.); stephanie.guidez@chu-poitiers.fr (S.G.); cecile.tomowiak@chu-poitiers.fr (C.T.); xavier.leleu@chu-poitiers.fr (X.L.); 2Department of Geriatric Medicine, Poitiers University Hospital, 2 Rue de la Milétrie, 86021 Poitiers, France; evelyne.liuu@chu-poitiers.fr; 3Department of Internal Medicine, Châtellerault Hospital Center, 1 rue du Dr Luc Montagnier, 86106 Châtellerault, France; thomas.systchenko@chu-poitiers.fr; 4Department of Nephrology, Poitiers University Hospital, 2 Rue de la Milétrie, 86021 Poitiers, France; vincent.javaugue@chu-poitiers.fr; 5Department of Rhumatology, Poitiers University Hospital, 2 Rue de la Milétrie, 86021 Poitiers, France; geraldine.durand@chu-poitiers.fr; 6Department of Internal Medicine, and CIC1402 INSERM unit, Poitiers University Hospital, 2 Rue de la Milétrie, 86021 Poitiers, France

**Keywords:** multiple myeloma, immunotherapy, transplant, novel agents, relapse

## Abstract

**Simple Summary:**

The therapeutics of multiple myeloma (MM) have greatly evolved in recent years both in the first-line setting and at relapse, and for both young and older patients. While still being considered as uncurable, MM survival has significantly increased. The development of immunotherapy (naïve and modern) largely contributed to these progresses, providing new effective drug combinations with acceptable toxicity profiles. Interestingly, some traditional concepts remain up to date, such as high-dose melphalan followed by autologous transplant, which can allow treatment intensification for eligible patients. Yet, with innovative response assessment techniques, allowing new response objectives, and treatment associations enabling profound responses, this, too, might be questioned in the near future. As the MM landscape is continuously moving, we sought to provide a review on the recent advances in the field of treatments in 2020.

**Abstract:**

The survival rate of multiple myeloma (MM) patients has drastically increased recently as a result of the wide treatment options now available. Younger patients truly benefit from these innovations as they can support more intensive treatment, such as autologous stem cell transplant or multiple drug association (triplet, quadruplet). The emergence of immunotherapy allowed new combinations principally based on monoclonal anti-CD38 antibodies for these patients. Still, the optimal induction treatment has not been found yet. While consolidation is still debated, maintenance treatment is now well acknowledged to prolong survival. Lenalidomide monotherapy is the only drug approved in that setting, but many innovations are expected. Older patients, now logically named not transplant-eligible, also took advantage of these breakthrough innovations as most of the recent drugs have a more acceptable safety profile than previous cytotoxic agents. For this heterogenous subgroup, geriatric assessment has become an essential tool to identify frail patients and provide tailored strategies. At relapse, options are now numerous, especially for patients who were not treated with lenalidomide, or not refractory at least. Concerning lenalidomide refractory patients, approved combinations are lacking, but many trials are ongoing to fill that space. Moreover, innovative therapeutics are increasingly being developed with modern immunotherapy, such as chimeric antigen receptor T-cells (CAR-T cells), bispecific antibodies, or antibody–drug conjugates. For now, these treatments are usually reserved to heavily pre-treated patients with a poor outcome. MM drug classes have tremendously extended from historical alkylating agents to current dominant associations with proteasome inhibitors, immunomodulatory agents, and monoclonal anti-CD38/anti SLAMF7 antibodies. Plus, in only a couple of years, several new classes will enter the MM armamentarium, such as cereblon E3 ligase modulators (CELMoDs), selective inhibitors of nuclear export, and peptide–drug conjugates. Among the questions that will need to be answered in the years to come is the position of these new treatments in the therapeutic strategy, as well as the role of minimal residual disease-driven strategies which will be a key issue to elucidate. Through this review, we chose to enumerate and comment on the most recent advances in MM therapeutics which have undergone major transformations over the past decade.

## 1. Introduction

Multiple myeloma (MM) is a plasma cell malignancy which represents nearly 10% of all hematological malignancies. MM evolves from a pre-malignant stage monoclonal gammopathy of undetermined significance (MGUS) and an asymptomatic intermediate stage called smoldering MM, before requiring treatment. The need for treatment is reached when there is evidence of one or more myeloma defining events (MDE)/end organ damage (EOD): the historical hypercalcemia, renal failure, anemia, or lytic bone lesions (CRAB) criteria, and recently, the sixty percent (60%) marrow involvement in plasma cells, light chain ratio >100, MRI >1 focal lesion (SLIM) criteria, which were elaborated to identify earlier patients with an immediate risk of developing MDE/EOD (80% at 2 years). 

Even though the treatment of newly diagnosed multiple myeloma (NDMM), transplant-eligible (TE), and not transplant-eligible (NTE) patients has remarkably improved over the years and multiple treatments are available, the disease will eventually relapse as MM remains not curable. Therefore, treating MM is a challenge, and finding the appropriate strategy to increase survival is the key issue. Still, from the old cytotoxic drugs to passive or active immunotherapy, the treatment of MM has greatly evolved over the past decade. Patients benefited from these innovations with prolonged survival, from a median of 3–5 years to 7–10 years nowadays. The MM armamentarium has never been so wide, which allows adequate strategies and optimal sequences of drugs for almost every patient, from the youngest and fittest to the elderly and frailest patients. Yet, progress is still awaited with novel immunotherapy and cellular therapies. 

In this review, we comment on the most interesting advances, in our opinion, in the treatment of MM, from transplant-eligible to not transplant-eligible and relapse and/or refractory MM patients.

## 2. Transplant-Eligible Patients

Novel agents have modified the multiple myeloma landscape since the vincristine-adriamycine-dexamethasone (VAd) regimen, continuously improving the depth of response of the induction phase, from doublet to recent quadruplet-based regimens, but also allowing a post-transplant therapy, from consolidation to maintenance. Overall, the survival of patients has significantly improved with first-line treatment and the progress developed at each step. However, the conditioning regimen of autologous transplantation has never benefited from these developments as it remains, 30 years later, based on the high dose of melphalan. In parallel to increased activity, combinations of drugs are now usually less toxic than previous treatments that were essentially based on cytotoxic drugs. Younger patients (usually ≤65 years old) have first and mostly benefited from the drug breakthrough innovations because they are eligible for more intensive strategies.

### 2.1. Induction Regimens

Finding the optimal induction regimen has been a long-time goal in MM. Induction is a crucial step in order to obtain a fast control of the disease with minimal toxicity. With the treatment options extending, multiple drug combinations are being investigated and approved.

Currently, the triplet induction regimen is the standard in the US and Europe, mostly represented by first-in-class proteasome inhibitor (PI) bortezomib-based triplet regimens, from bortezomib, cyclophosphamide, and dexamethasone (VCd) to bortezomib, thalidomide, and dexamethasone (VTd), and recently, bortezomib, lenalidomide, and dexamethasone (VRd) (Figure 1). The most commonly used regimen remains VTd for induction (32%), but mostly in Europe [1]. VTd was found superior to VCd in terms of very good partial response (VGPR) rates (VTd 66.7% vs. 56.2% VCd, *p* = 0.05). However, no direct comparison between VTd and VRd has ever been reported. However, an integrated analysis of four randomized controlled trials (GEM2005, GEM2012, IFM 2009, IFM 2013-04) showed an improvement in ≥VGPR rate with VRd versus VTd, and VRd also led to less peripheral neuropathy and treatment related adverse events (AE) [2]. Thus, depth of response appears higher with VRd but the impact on survival is not warranted; in fact, there are no comparative data about progression-free survival (PFS) and overall survival (OS) between VTd and VRd. Moreover, there is room for improving the already existing induction regimens and quadruplet regimen (traditional triplet + monoclonal antibody) which has expanded in clinical studies. The CASSIOPEIA study was one of the first to test daratumumab, an anti-CD38 monoclonal antibody (mAb)-based induction treatment, by combining it to VTd [3]. The results showed complete response or better (≥CR) rate of 39% in the dara-VTd group vs. 26% in the VTd alone group (*p* < 0.0001), and minimal residual disease (MRD) 10^−5^ negativity rate was also in favor of dara-VTd, 64% vs. 44% (*p* < 0.0001), respectively. In the phase 2 GRIFFIN study, dara-VRd vs. VRd, primary endpoint stringent CR (sCR) by the end of post-autologous stem cell transplant (ASCT) consolidation, the results were in favor of dara-VRd 42.4% vs. 32.0% for VRd (*p* = 0.068), and rates of MRD 10^−5^ negativity largely benefited dara-VRd 51.0% over VRd 20.4% (*p* < 0.0001). The phase 3 PERSEUS (dara-VRd vs. VRd, NCT03710603) trial will likely confirm these results and we can reasonably expect that they might be even better than CASSIOPEIA. Several other clinical trials testing quadruplets as an induction regimen are ongoing with different combinations of drugs and different mAbs; elotuzumab-VRd in the GMMG-HD6 trial (NCT02495922), elo-KRd in the DSMM XVII trial (eudraCT 2017-001616-11), or isatuximab-VRd in the GMMG-HD7 trial (NCT03617731).

Among the key questions which remain to be answered regarding induction therapy is whether the minimal residual disease (MRD)-driven tailored treatment prolongation, including ASCT, can find its place in daily practice. The phase 2 MASTER trial [4] proposed to treat patients with all existing agents, quadruplet regimen of daratumumab, carfilzomib, lenalidomide, and dexamethasone (dara-KRd) for four cycles, followed by ASCT and MRD-based dara-KRd consolidation) until they achieved two consecutive MRD 10^-5^ negative responses. MRD was assessed by using next generation sequencing (NGS) at the end of induction, after ASCT, and after four cycles of consolidation which could be repeated for another series of consolidation. MRD-negative remission rate was 40% after induction, 73% post-transplant, and 82% during consolidation. This study is the first amongst many to investigate the ability to stop treatment based on MRD for patients, which seems to validate a sort of ‘good-risk profile’ by entering MRD negativity at a deep level. 

Although quadruplet regimens will surely be extensively used in the future, induction regimen also depends on economic factors, country restrictions, and authority approval, and so VTd and VRd induction regimens might still be seen as the gold standards for transplant-eligible NDMM patients for a couple of years. Novel immunotherapy-based treatments with the modern chimeric antigen receptor T-cell (CAR-T) or bispecific antibody-based immunotherapy are on the verge of starting, and it will take some time and more data before they can replace or complement current first-line treatments, replace ASCT, or lead more patients to MRD negativity and allow early treatment discontinuation.

### 2.2. Autologous Stem Cell Transplant

High-dose chemotherapy plus autologous stem cell transplant (ASCT) represents the most intensive care approach in MM nowadays (allo-SCT is becoming very rare for MM). Several studies, even recently with optimal induction regimens, have repeatedly demonstrated that ASCT was validated as one of the most effective debulking frontline treatments for young or fit patients demonstrating better event free survival (EFS)and OS. The Intergroupe Francophone du Myélome (IFM) 2009 study is one of the most recent studies to raise the question of ASCT in the context of one of the most effective induction regimens, VRd [5]. The median PFS was significantly longer in the group that underwent ASCT (50 months vs. 36 months, *p* < 0.001). This benefit could be seen across all subgroups of patients, and responses were also deeper in the transplant group with a higher CR rate and MRD negativity rate. However, there was no difference in OS between the two groups (82% vs. 81%), leaving open the question of transplant upfront versus delayed at relapse, and also that, potentially, specific subgroups, yet to be identified with sufficient statistical power, might not benefit more from transplant versus non-transplant. Interestingly, patients who reach to MRD negativity remains questionable to a transplant approach. The FORTE trial confirmed the important role of ASCT in order to achieve deep responses while using a modern triplet combination of carfilzomib, lenalidomide, and dexamethasone (KRd) but did not really clarify the role of ASCT to achieve higher rates of MRD negativity [6]. Indeed, the risk of early relapse was significantly reduced in the KRd-ASCT-KRd arm, 8% vs. 17% (*p* = 0.015), but rates of MRD negativity were similar between the two arms, KRd-ASCT-KRd 58% vs. 54% KRd for 12 cyles. So far, no data about survival have been reported. Even if the rate of MRD negativity was similar between the two arms, sustained MRD at 1 year was higher for KRd-ASCT (90% vs. 78%), and, therefore, one could expect that with a longer follow-up, we might see a difference in terms of survival. Moreover, this trial has demonstrated once more that the triplet cyclophosphamide-based combination with PI (KCd) was inferior to the immunomodulatory agent (IMiD) + PI-based combination. 

Therefore, to date, early ASCT still remains the standard of care for this population in the initial treatment plan, and every eligible patient should be considered for ASCT [7]. Future studies might challenge this approach by investigating triplet and quadruplet regimens in addition to anti-CD38 mAbs as an alternative to ASCT. Indeed, ongoing trials are looking into ASCT versus no ASCT based on MRD status at the end of induction for non-high risk NDMM patients, considered as the most important predictive prognostic factor for this group. ASCT could then be possibly delayed and used as a salvage therapy or at relapse. Other potential specific subgroups of patients might also benefit from a delayed or no ASCT approach. The importance of MRD negativity will surely grow and might become a surrogate endpoint for PFS/OS and challenge the “systematic” use of ASCT.

### 2.3. Post-Transplant Consolidation

The results of the phase 3 staMINA trial, a three-arm study (induction + ASCT, induction/ASCT/four cycles of VRD consolidation, tandem-ASCT), did not show any additional benefit with more treatments after ASCT, such as consolidation therapy or a second ASCT, while the three arms were followed by maintenance [8]. With a median follow-up of 38 months, the median PFS was not significantly improved in patients in the consolidation or tandem arms compared with maintenance directly after ASCT (*p* = 0.37). Rates of OS were also similar between the three groups: 82%, 85.7%, and 83%, respectively. Recently, the updated analysis with a 6-year follow-up confirmed these results with similar rates of OS: 76.4% (Auto/len maintenance); 74.9% (Auto/VRD); 73.1% (Auto/Auto), (*p* = 0.8). Yet, high-risk patients benefited from the tandem-ASCT with a 6-year PFS of 43.6% vs. 26% for the maintenance group (*p* = 0.03). On the contrary, the phase 3 EMN-02 trial supported the use of consolidation therapy with VRd after ASCT; median follow-up of 42.1 months, median PFS 58.9 months with consolidation vs. 45.5 months with no consolidation (*p* = 0·014). Post-transplant consolidation is, therefore, still debated. 

### 2.4. Maintenance Therapy

As there is still no curative treatment for MM and all the patients will eventually relapse, prolonged treatments post-ASCT are now well established. Maintenance therapy prolongs the duration of response and increases median PFS/OS, and even, in certain cases, deepens this response. Lenalidomide is currently the only drug approved in this indication.

In 2017, a meta-analysis of three randomized controlled trials (CALGB100104, RV-MM-PI-209, IFM2005-02) [9] showed a statistically significant increase in PFS with lenalidomide maintenance between the two groups: 52.8 months and 23.5 months (IC 95%: 0.41–0.55), respectively. The benefit was also seen in terms of OS with median OS not reached with lenalidomide maintenance and 86 months for the observation group (*p* = 0.001). Overall, a 25% reduction in the risk of death could be noticed, as well as an increase of 2.4 years in median survival. The most recent phase 3 study MYELOMA XI [10] confirmed a significant advantage of lenalidomide with a median PFS reaching 57 months compared to 30 months (*p* = 0.0001), and an OS benefit with a 3-year OS of 87.5 months vs. 80.2 months (*p* = 0.014), respectively. Currently, lenalidomide is mainly given until progression to maximize both PFS and OS, even though there has never been a direct comparison between fixed and continuous R maintenance.

Obviously, new drugs are being investigated for maintenance therapy in order to improve survival in TE patients. The TOURMALINE-MM3 study [11] introduced ixazomib, a novel oral PI agent, as a maintenance drug. The median PFS was 26.5 months vs. 21.3 months (*p* = 0.0023), respectively. Ixazomib is, theoretically, a good candidate for maintenance treatment because of its easy (3 times/28 days uptake) oral administration and acceptable safety profile; however, the results in terms of PFS appear inferior to lenalidomide. The CASSIOPEIA trial explored daratumumab maintenance monotherapy for 2 years, and the results are awaited.

So far, maintenance treatments were given as a monotherapy in order to prevent toxicity on a long-term basis, but with the advent of treatments which are less and less toxic, one could imagine that bi- or tri-therapy could be given after ASCT and consolidation, still with the objective to deepen responses and convert more patients to an MRD-negative status. For example, the maintenance drug was carfilzomib and lenalidomide (KR) versus lenalidomide alone (R) in the FORTE trial. Similarly, the PERSEUS study (NCT03710603) will test maintenance with daratumumab and lenalidomide in comparison to R alone and will question the role of MRD to stop treatment during maintenance. In addition, we can also report the GMMG-HD6 trial which is evaluating maintenance with anti-SLAMF7 elotuzumab in association to lenalidomide versus lenalidomide alone (NCT02495922). Another interesting trial is the ongoing EMN18 exploring ixazomib-daratumumab maintenance versus ixazomib (NCT03896737).

One of the next steps will probably be to evaluate the choice of drug and duration of maintenance based on MRD status. We will have to wait for the results of clinical trials, such as the upcoming IFM2020 or the SWOG1803/DRAMMATIC study (NCT04071457), to answer many of these questions.

### 2.5. Risk-Adapted Therapy

Another important issue for the future of MM therapy is surely the risk-adapted strategy. High-risk (HR) patients and standard-risk (SR) patients should probably not be treated the same way. For the SR subgroup, the tendency would be to treat patients less intensively and/or for a shorter period. Studies are willing to demonstrate that ASCT could surely be questioned in patients who would reach an MRD-negative status at the end of induction. Yet, the risk here is clearly to undertreat SR patients. Similarly, in this subset, the need for a quadruplet induction regimen is also questionable, as well as the type and duration of post-transplant consolidation and maintenance treatments. Concerning HR patients, the issue is completely different given the difficulty for most HR patients to achieve an MRD-negative status. Moreover, it also appears that the treatment of HR patients in the relapse setting is extremely disappointing as well. Therefore, multiple studies enrolling HR patients are trying an “all-in” approach. The GMMG-CONCEPT trial (NCT03104842) brought the concept of treating patients with a quadruplet treatment based on their HR status (defined by the presence of del17p or t(4;14) or t(14;16) or > 3 copies 1q21 and ISS 2 or 3 stage disease). Isatuximab-KRd was given for six cycles of induction (+2 if no ASCT), four post-ASCT consolidations, and isa-KR maintenance. Combining nearly all the drugs available in MM could probably be an interesting option for HR patients who have a poor outcome. The risk-based approach in MM is only starting and it might take more time to define which treatment regimen is more suited for which subset of patients, standard-risk or high-risk patients. Furthermore, in the future, we might potentially identify other risk-based subgroups (t(11;14), PET-CT negative, etc.)

## 3. Non-Transplant Eligible Patients 

The median age of MM patients at diagnosis is nearly 70 years; consequently, most of the patients diagnosed with myeloma are considered as old. The complexity of treating this subset of patients principally lies in the impossibility for most of them to undergo an intensive treatment, such as ASCT, and therefore to effectively debulk the bone marrow. Non-transplant eligible (NTE) patients would, therefore, have a poorer outcome than younger ones. However, through the years, innovations in the MM field and especially the emergence of immunotherapy offered new options for this group with effective and relatively non-toxic drugs.

### 3.1. Heterogeneous Patients, Different Perspectives

The treatment selection in NTE patients can no longer solely be based upon chronological age or performance status [12]. It is now well-admitted that these two criteria are not sufficient to describe this entire population with marked heterogeneity. The approach for treating such patients has to, therefore, be more personalized, and the objectives can really differ from one patient to another. The heterogeneity inside this subgroup is significant, with patients presenting great variations in their inherent characteristics (age, comorbidities, functional status, physiological reserve) and environmental factors (access to care, social support). Geriatric assessment is therefore relevant to distinguish patients and guide the strategy. Detecting frailty has become essential to determine the right treatment plan and not overtreat or undertreat patients. Various assessment tools were developed and validated for older patients: the Myeloma Working Group frailty score (IMWG) [13], the Revised Myeloma Comorbidity Index (R-MCI) [14], the Hematology Oncology Frailty (HOF) score [15], or the UK Myeloma Research Alliance Risk Profile (MRP) [16].

Currently, fit patients could be eligible for a triplet first-line treatment, with a PI and an IMiD, possibly followed by ASCT since it remains the best option to reach for deep responses, and it is offered in a small number of centers for these patients [17]. This strategy is inspired by and similar to the treatment plan of younger patients (Figure 1). The aim would be, therefore, to obtain the best possible response—probably sustained MRD negativity. For intermediate-fit or unfit patients, triplet regimen is probably well-suited too, but they can definitely not tolerate ASCT, and management of the drugs has to be very careful in order to limit toxicities. For this subset, the balance between efficacy and safety is essential. Indeed, too many dose reductions could negatively impact their response and their outcome. Finally, for the frailest patients, doublet regimens, PI- or IMiD-based (bortezomib-dex, Vd, or lenalidomide-dex, Rd), are more acceptable. For these particular patients with increased comorbidities and limited life expectancy, safety is the key issue over treatment response, including the best supportive care as a major part of therapy management. Early discontinuation of dexamethasone is an important objective of ongoing studies for this population.

### 3.2. Triplet Melphalan-Based or Double/Triplet Lenalidomide-Based Therapies 

Historically, the interest in NTE patients was a little restricted in comparison to younger patients, with fewer clinical trials specifically designed for them, but this growing population is gaining more attention as current drug development is providing appropriate and tolerable options.

Immunotherapy (IT) already represents a massive change in the MM scene, with the anti-CD38 mAb ahead. Expectedly, daratumumab was largely, and successfully, tested for NTE patients across multiple clinical trials.

Presently, the standards of care for NTE patients are surely melphalan-prednisone-bortezomib (MPV), Rd, and/or VRd. It should also be noted that the combination of bortezomib, cyclophosphamide, and dexamthesone (CyBorD/VCd), even if less used nowadays, is still an acceptable regimen, though the “real” CyBorD regimen with bortezomib 1.5 mg/m² is not available in many countries. Melphalan-based associations are progressively being outrun by IMiD combinations, but fixed-duration MPV, which was brought by the VISTA trial, is an effective treatment and is available in most countries [18]. However, MPV has shown some significant toxicity, particularly neurological, with a higher risk of peripheral neuropathy and the risk of developing myelodysplasia, which comes along with the long-term utilization of melphalan. Furthermore, the median PFS of almost all types of MPV regimens hardly increased beyond 18 months. The phase 3 ALCYONE study has therefore investigated the combination of MPV plus daratumumab [19]. The most recent analysis demonstrated an important improvement in terms of PFS; median PFS 36.4 months for dara-MPV [20]. Regarding MRD negativity rate, the results favored the use of dara-MPV with 23.3% MRD-negative status compared to 6.2% in the control group. Still, MPV might disappear as a backbone standard of care.

The emergence of lenalidomide was, thus, a major step forward for NTE patients with its easy all-oral administration and very acceptable safety profile. Rd first showed superiority over a combination of melphalan, thalidomide, and dexamethasone (MPT) in the FIRST study (median PFS of Rd continuous 26 months vs. 21.9 months, *p* < 0.0001, respectively) [21]. Rd also offers a viable option in the long run for most patients, even the frailest. Nevertheless, the control of MM could be improved in the context of a triplet-based combination with bortezomib; SWOG S0777 trial (VRd vs. Rd) [22]. The ≥CR rate was 15.7% for VRd vs. 8.4%. VRd also increased the toxicity signature with more grade 3 or higher adverse events (AE); 82% vs. 75%. Furthermore, we moved from fixed-duration therapy (MPV) to a prolonged treatment approach with Rd-based associations for NTE patients upfront in order to improve the outcome, achieve deep responses, and delay relapse [10,21,23]. The addition of carfilzomib to Rd was also tested for NTE patients in the phase 3 ENDURANCE trial (KRd vs. VRd), but it did not improve the progression-free survival and added more toxicity (median PFS 34.6 months for KRd vs. 34.4 months for the VRd group, *p* = 0.74) [24]. Nevertheless, it confirmed that VRd is suitable and efficient for older patients.

The results of the phase 3 MAIA trial [23] showed dara-Rd vs. Rd, both until progression, should completely change the first-line NTE landscape in the years to come. The 36-month PFS was 68% for dara-Rd vs. 46% for the control group [25] (HR 0.56, CI95% 0.43–0.73). Again, the MRD negativity rate was impressive, with 24.2% of the patients having an MRD 10^−5^ negative status with dara-Rd.

Following this, the phase 3 CEPHEUS trial will evaluate the safety and efficacy of a quadruplet regimen for NTE patients with dara-VRd vs. VRd (NCT03652064), as well as the IMROZ study, with isatuximab-VRd vs. VRd (NCT03275285). Daratumumab is not the only anti-CD38 mAb; isatuximab, with a slightly different mechanism of action, now represents an alternative. Daratumumab is increasingly being tested for older patients and will be associated with many other MM drugs, such as ixazomib and no dexamethasone in the IDARA trial (NCT03652064), and in the phase 2 HOVON 143 trial (ixazomib, daratumumab, and low dose dexamethasone) for frail or unfit patients (EudraCT 2016-002600-90), or even with ixazomib, lenalidomide, and dexamethasone in a phase 2 trial (NCT0400909). The question of the added value of bortezomib in the anti-CD38-Rd backbone is questioned in the next IFM2020-05/BENEFIT study. 

The addition of the anti-CD38 mAb has clearly transformed the treatment combinations and paradigm in NTE MM patients, including the frail ones, and allows continuous treatment. 

### 3.3. Ixazomib: A Convenient Drug for NTE Patients?

In addition, ixazomib, with its convenient oral administration, appears to be an interesting choice for transplant-ineligible patients and is currently being explored in a few clinical trials challenging Rd as a backbone standard of care or for continuous/maintenance therapy. The phase 3 TOURMALINE-MM2 trial has investigated ixa-Rd vs. Rd for newly diagnosed NTE patients but the first results did not show any significant improvement in terms of PFS: ixa-Rd 35.3 months vs. 21.8 months for the Rd group (*p* = 0.073) (Takeda press release, 2020). Besides, the UK FiTNEss trial (NCT03720041) is also exploring the association of ixazomib, lenalidomide, and dexamethasone but in a frailty-adjusted strategy. All the participants will receive 12 cycles of ixa-Rd standard dose or dose-adjusted in function of their frailty status, followed by ixa-R maintenance or R alone, still depending on their frailty features. On the other hand, the TOURMALINE-MM4 study (NCT02312258) is evaluating ixazomib maintenance against a placebo after initial therapy for NTE patients.

## 4. Relapse or Refractory Multiple Myeloma 

Most of the new drugs are initially tested in the relapse setting before entering the front-line treatments. Thus, relapse or refractory MM (RRMM) has also undergone a profound transformation. Novel generation IMiDs (lenalidomide, pomalidomide) and PI (carfilzomib, ixazomib) are at disposal, as well as mAbs (daratumumab, isatuximab, elotuzumab), along with a series of new drugs and families with new mechanisms of action (melflufen, selinexor, venetoclax). Certainly, the modern ‘armed’ immunotherapies, such as bispecific antibodies and CAR-Ts as well as cereblon E3 ligase modulators (CELMoDs) (CC-220 and CC-92480) and conjugated immunotherapies, represent the greatest hope for the future in MM. Still, with such a large selection of treatments, several questions arise as to which is the better option and for who.

### 4.1. First Relapse

The majority of patients will be treated with a triplet regimen including a PI and an IMiD lenalidomide as the front-line treatment in the future (Figure 2). The challenge is to find an adequate and effective combination without including any refractory drugs, and ideally keeping, for as long as possible, the patients under treatments that the tumor cells are naïve from. 

**Not IMiD/lenalidomide refractory** (progression under treatment or <60 days after the end of treatment). A triplet association with lenalidomide is probably the better option. To date, as still only few patients would have received daratumumab, the triplet dara-Rd is a valid choice based on the data of the phase 3 POLLUX trial, with an impressive median PFS of 45.8 months for dara-Rd vs. 17.5 month for Rd (*p* < 0.0001) [26], and a ≥CR rate of 57% vs. 24%, respectively. Interestingly, dara-Rd also performed well for patients exposed to lenalidomide, although it concerned a quite small cohort of patients (*n* = 50). Another compelling option is carfilzomib associated to Rd (KRd) as introduced in the ASPIRE trial. The median PFS with KRd was 26.3 months vs. 17.6 months (*p* = 0.0001) [27], and the median OS in the final analysis was 48.3 months for KRd versus 40.4 months for Rd (*p* = 0.0045) [28]. Other possible options include combinations such as elotuzumab-Rd and ixazomib-Rd; median-PFS 19.4 months in the ELOQUENT-2 trial (*p* = 0.001) [29] and 20.6 months in the phase 3 TOURMALINE-MM1 trial (*p* = 0.0012) [30]. Indeed, we will not mention, in this review, Rd, RCd, VRD, etc., which are obviously valid options but mostly used in first-line responses.

**Lenalidomide refractory patients.** Lenalidomide (R) refractory patients are now becoming a growing population since R is increasingly used in the first-line setting. The choice is a little limited and potentially suboptimal nowadays since most efforts focused on R combinations. However, new studies are populated with R-sparing regimens in order to address this issue. The first two studies that are non-R containing, but with a few R-refractory patients, are the CASTOR trial (dara-Vd vs. Vd); median PFS of 16.7 months with dara-Vd, (*p* < 0.0001) [31]. In the lenalidomide refractory subgroup analysis, dara-Vd was also significantly more effective; median PFS 9.3 months vs. 4.4 months (*p* = 0.0002), although the PFS decreased significantly. Using carfilzomib in this context will be limited, for now, to the doublet association Kd as it was validated in the ENDEAVOUR trial [32]—the median PFS for Kd (K 56 mg/m² Q2W) was 18.7 months (*p* < 0.0001) vs. 9.4 months with Vd (large number of patients exposed to V, all K naïve).

Multiple trials are expected to bring more treatment options to the field with more data on R-refractory RRMM. The triplet carfilzomib-based options start with CANDOR (dara-Kd vs. Kd), median PFS which was not reached vs. 15.8 months with Kd (*p* = 0.0014) [33], and IKEMA (isa-Kd versus Kd), median PFS not reached for Isa-Kd vs. 19.15 months for Kd (*p* = 0.0007). The results in the R-refractory group are yet to be shown but will be key to compare the treatment options. The statement here is that dara/anti-CD38-Kd is already better than Kd and demonstrates that triplet-based IMiD-free regimens can also be developed in early-relapsed MM. Other K-combinations might be developed, although maybe not for approval, such as K + pomalidomide, K + cyclophosphamide, K + venetoclax, etc.

One of the major improvements in this setting is the emergence of triplet pomalidomide-based (third generation IMiD agent) regimens. Pomalidomide appears to be the backbone of many drug combinations after the first relapse, including for R-refractory patients. In real-world clinical practice, as well as in most clinical trials, pomalidomide can be administered to patients who received at least two prior therapies including a PI and lenalidomide and are refractory to the last line of treatment. The triplet pomalidomide-based options have been first studied with the phase 3 OPTIMISMM study (poma-Vd vs. Vd), the median PFS was 11.2 months for poma-Vd vs. 7.10 months for Vd (*p* < 0·0001). Importantly, for patients with only one previous line of treatment, the median PFS was longer (20.73 months, *p* = 0.0027), which could also be seen for lenalidomide refractory patients (median PFS 17.84 months, *p* = 0.03). Another important combination with pomalidomide was validated in the phase 3 ICARIA study (poma-dex vs. isatux-poma-dex) [34], which demonstrated that isatuximab can be an interesting drug partner for pomalidomide, including for patients with advanced disease. The median PFS was 11.5 months for isa-Pd, for a median of four prior lines of treatment, vs. 6.5 months with Pd (*p* = 0.001); in addition, we noted 5.2% of patients with MRD 10^−5^ negativity. Similar results are provided with daratumumab in the phase 1b EQUULEUS study [35], but we will have to wait for the results of the phase 3 APPOLO study to conclude (NCT03180736). Furthermore, pomalidomide was also investigated with elotuzumab in the phase 2 ELOQUENT-3 study [36] with encouraging median PFS in advanced RRMM. 

### 4.2. Promising Drugs for Heavily Pre-Treated Patients

#### 4.2.1. Targeted Therapy: Selinexor, Melflufen, etc.

Another potential target for MM treatment could be the nuclear export protein XPO1 which is overexpressed by MM cells and regulates cell survival and intracellular protein transport. Selinexor is a first-in-class oral XPO1 inhibitor and has already been investigated in clinical trials. The phase 2b STORM trial showed an overall response rate (ORR) of 21% with selinexor (80 mg × 2/week, 4-week cycles) in combination with dexamethasone, and median PFS was 2.3 months in heavily pre-treated patients [37]. The other phase 2 STOMP trial evaluated selinexor with bortezomib and dex and the results demonstrated an ORR of 63% and median PFS of 9 months for the association [38]. The phase 3 study BOSTON (NCT0311056262), selinexor-Vd vs. Vd, should confirm these data and help to learn more on the safety profile, particularly of gastrointestinal disorders, so that we can improve quality of life with Selinexor.

Melflufen (melphalan flufenamide), a first-in-class peptide-conjugated alkylator, has emerged as a new potential treatment alternative for RRMM patients. Melflufen is metabolized into melphalan and p-fluorophenylalanine via aminopeptidase-dependent cleavage inside myeloma cells. The melphalan metabolite then accumulates into MM cells which confers its anti-myeloma toxicity. The cytotoxicity of melflufen overcame melphalan resistance in vitro and in xenograft models. The results of the phase 1-2 study showed an MTD of 40 mg in association with dexamethasone [39]. Patients with combination therapy had an ORR of 31%; in the single agent cohort, the ORR was 8%. The most common grade 3–4 AEs were thrombocytopenia (62%) and neutropenia (58%), while non-hematological toxicity was rare. The results of the phase 2 HORIZON (melflufen 40 mg IV/28 days + dex 40 mg/week, 20 mg if >75 yo) trial for RRMM (≥2 prior lines of treatment, including a PI and IMiD, and refractory to pomalidomide and/or anti-CD38) demonstrated an ORR of 29% and median PFS was 4.2 months (3.7-4.9). Furthermore, among triple-class refractory patients, the ORR was 24% and median PFS was 3.9 months. The phase 3 OCEAN (NCT03151811) trial, melfufen-dex vs. poma-dex, which is ongoing, will surely clarify the outcome with melflufen in association to dexamethasone.

#### 4.2.2. T(11;14): A Subgroup for Tailored Therapy

**Antibcl2 targeting.** Another potential option for patients suffering from advanced MM is the use of venetoclax. This first-in-class anti-BCL2 inhibitor is utilized in several other hematological malignancies, such as chronic lymphocytic leukemia (CLL) or acute myeloid leukemia (AML). BCL2 is an anti-apoptotic protein that promotes cell survival. MM cells harboring the t(11;14) translocation usually express a higher ratio of BCL2 and, therefore, an increased sensitivity to venetoclax. One of the resistance mechanisms of anti-bcl2 agent is the increase in expression of mcl1, therefore the association of venetoclax to PI agents was a potential way to overcome this mechanism. Venetoclax was then tested with bortezomib in a promising phase 1b followed by the phase 3 BELLINI trial which analyzed the efficacy of venetoclax, bortezomib, and dexamethasone (ven-Vd) in comparison to bortezomib and dexamethasone (Vd) for early RRMM patients, with a specific subgroup analysis for t(11;14) patients [40]. The study provided impressive results for the patients with t(11;14) which had a better outcome (median PFS not reached vs. 9.5 months, *p* = 0.002) with less treatment-related toxicity. Future use of venetoclax in RRMM will possibly be reserved to the subgroup with t(11;14), and thus represents the first biomarker-based treatment in MM.

#### 4.2.3. Modern Immunotherapy

**CAR-T cells.** CAR-T cells are finding their way into most hematological malignancies, starting from acute lymphoblastic leukemia to MM. In the MM field, the utilization of CAR-T cells lies in second generation CAR (with a costimulatory molecule, 4-1bb or CD28) and they principally target B cell maturation antigen (BCMA), which is very specific to the plasma cell (PC), and participate in the differentiation of B cells into PCs and promote their survival. For now, the two most advanced CARs are ide-cel/bb2121 (and its modified version with a PI3K inhibitor, bb2121-7) and JNJ4528 (previously named LCAR-B38M). The first results with bb2121, from the phase 1 study CRB 401, showed an ORR of 85% with 45% CR in end-stage RRMM patients and a median PFS of 11.8 months [41]. These results led to the development of a phase 2 study, KarMMa1 [42]; most of the patients were heavily pre-treated (median of six previous treatments) and 84% were triple-class (IMiD, PI, anti-CD38 mAb) refractory. The ORR was 73%, including 33% CR, and median PFS was 8.8 months. Other promising data come from the LEGEND-2 and CARTITUDE-1 studies using the CAR JNJ4528/LCARB38M. This CAR also has a 4-1bb co-stimulatory molecule but is structurally different from ide-cel because it has two BCMA-targeting domains, which could potentially represent an improvement in terms of efficacy to target clones with low BCMA expression. The CARTITUDE-1 study was designed as a confirmation in the US of the phase 1 LEGEND-2 study [43] with identical CAR-T renamed JNJ4528 [44]. The ORR was 100%, with 69% achieving ≥CR and 86% achieving VGPR or better. Even though CAR-T cell therapy may lead to deep responses, the response durability remains one of the major issue, and thus phase 2 and 3 studies will be needed to better establish the place of these drugs in the MM treatment strategy.

For all CAR-T constructs, the main problem remains the early safety profile and, especially, the cytokine-releasing syndrome (CRS). Nearly all patients are experiencing CRS—76% of patients in CRB-401, 90% in LEGEND-2, and 93% in CARTITUDE 1. Most of the CRS diagnoses were grade 1-2—70%, 83%, and 80%, respectively. CRS usually happens very soon after CAR-T infusion with a median of 2 days in the CRB-401 study, 9 days in the LEGEND-2 study, and 7 days in the CARTITUDE-1 study. Moreover, clinicians are progressively learning how to manage CRS, and it should soon become routine practice in medical centers performing CAR-T cell treatments. CAR-T cells are highly being investigated in MM, and we could move from autologous CAR-T cells to allogenic CAR-T cells, which could diminish the fabrication time and provide an ‘off-the-shelf’ therapeutic. Furthermore, CAR-NK could be an alternative to CAR-T cells in the future [45].

**Conjugated MAbs.** Belantamab-mafodotin, an antibody–drug conjugate, also represents an innovative and promising option for RRMM. The results of the phase 2 DREAMM-2 study (following the first in-human DREAMM 1 study) were encouraging and introduced belantamab as an effective option for patients who have been treated with all the conventional MM drug classes [46]. The ORR in DREAMM-2 with belantamab at 3.4 mg/kg IV every 3 weeks or 2.5 mg/kg as a single agent achieved was 31% patients in the 2.5 cohort and 34% in the 3.4 cohort. The safety profile was marked by corneal keratopathy-based toxicity with 27% and 21% of keratopathy in each group, respectively, and a hematological toxicity. Several DREAMM studies are being conducted or planned, such as the 2 phase 3 trials DREAMM-3 (belantamab vs. pomalidomide-dex) (NCT04162210), and DREAMM-9 (NCT04091126) which will test belantamab in association with VRd versus VRd alone in first-line setting for NTE patients. Belantamab is certainly a viable option for patients previously treated with IMiDs, PI, and anti-CD38 mAb.

**Bispecific MAbs.** Early data are also massively coming from new therapies which could later take place in the MM treatment strategy. Among them, bispecific antibodies, principally BiTEs (bispecific T cell engagers), which link functionally active T cells via CD3 and plasma/tumor cells in order to induce T cell activation and cause lysis of the antigen-expressing target cell, represent a very interesting innovation. The first proof of concept was provided by AMG 420, which binds BCMA on PCs and CD3 on T cells. The phase 1 dose-escalation study showed that at the maximum tolerated dose (MTD) of 400 µg/day (continuous infusions for 6-week cycles, up to 10 cycles), five MRD-negative CRs were obtained, as well as one VGPR and one PR, with a median duration of response of 9 months in RRMM with median of four prior lines of treatments [47]. Concerning the safety profile, 45% of the patients experienced a serious AE, 38% had a CRS, and two patients died of infections (aspergillosis and hepatitis fulminans related to adenovirus infection).

Other anti-BCMA bispecifics are in development, such as teclistamab, whose preliminary results for the first in-human phase 1 dose-escalation study were reported in patients with a median of six prior lines of treatment, and the ORR was 67% at the highest dosage. The most common AEs were mainly CRS, anemia, neutropenia, thrombocytopenia, and pyrexia. Bispecifics in clinical development are principally targeting BCMA, but other targets are being explored, such as the GPRC5D protein with an anti GPR5D/CD3 bispecific, which showed clinical in vitro activity for MM [48].

**CELMoDs.** Cereblon E3 ligase modulators (CELMoDs) represent a novel approach in the treatment of RRMM. IMiDs, such as lenalidomide or pomalidomide, mediate their anti-myeloma activity and, particularly, immunomodulatory activity via cereblon; CELMoDs, therefore, are novel and functionally different cereblon-targeting agents which lead to myeloma cell apoptosis and stimulate the immune system, and they could help overcome resistance to certain drugs. Iberdomide (CC-220) showed an ORR of 31% in association with dexamethasone in the first in-human phase 1b/2a multi-center dose-escalation (0.3–1.2 mg 21 days/28) study. Grade 3-4 AEs were reported for 72% of patients, mainly hematological toxicity. The population was heavily refractory to MM drugs (median prior regimens = 5). One of the other major CELMoDs currently in development is CC-92480 which is being investigated in a phase 1 study (NCT03374085). In preclinical studies, CC-92480 demonstrated antiproliferative effects and enhanced immune stimulatory effects for sensitive and resistant MM cells, possibly superior to CC220. This led the field to believe that CELMoDs will also play a key role in MM in the near future.

### 4.3. Checkpoint Inhibitors: An Unclear Role for Now

**Checkpoint inhibitors.** Checkpoint inhibitors are already used in several solid tumors (melanoma, lung cancer, etc.) and hematological malignancies (especially Hodgkin lymphoma), and were therefore studied in MM. One of the most important series of studies, the KEYNOTE trials program, investigated anti-PD1 pembrolizumab in association with IMiDs [49,50]. The development in MM was suspended due to high toxicity and an increased incidence of death. Other developments were studied in MM; for example, checkpoint inhibitors were tested with the hope to counter the resistance mechanisms of immunotherapy developed by plasma cells overexpressing immune checkpoint molecules. There is, thus, no clear role for checkpoint inhibitors in MM, yet.

## 5. Conclusion

The treatment of multiple myeloma has profoundly transformed over the years with novel families of drugs, novel mechanisms of action, and the development of tailored therapy and biomarker-based therapy, as well as the emergence of immunotherapy, naïve and modern, cellular, and antibody-based. Consequently, we are observing a paradigm shift in the way we treat MM, either upfront for NDMM, and at relapse. These developments have been accompanied with better safety profiles and reduced use of cytotoxic drugs, and the increased utilization of immunotherapy-based treatments has pioneered this endeavor. Expectedly, the quality of life of patients is constantly improving, despite being far more treated nowadays compared to the years before, along with a significant effort to improve supportive care. Interestingly, the definition of MM and the criteria for treatment have also evolved in parallel to treatments. 

Importantly, the understanding of the role of MRD negativity status, at least at 10^−5^, has profoundly transformed the regimens, sequence, and treatment approach globally in MM, and will certainly be the backbone for future changes in years to come. Certain subgroups clearly remain a challenge, namely the high-risk MM patients to whom negative MRD status appears to be one of the best markers of improved success and prolonged survival, but also the frailest patients that benefit from better geriatric assessment to allow careful identification of comorbidities and, therefore, the best treatment approach to limit the safety consequences of the treatments. 

The prospects are great in MM treatments, and while a “cure” remains a dream, functional cure/chronic MM is already there for a growing number of patients. Still, one can anticipate some limitations that will have to be addressed: (i) the development of quadruplet regimens early in the disease course, and particularly upfront—in other words, piling drugs in the first-line setting which could jeopardize the treatment options at relapse; (ii) most of the drugs used are meant to be given in a continuous way, but real-life experience has repeatedly demonstrated that continuous treatment is extremely harassing to patients either physically/cognitively or even psychologically, and lessening the treatment dose density/intensity can lead to early developments of relapse and mechanisms of resistance. Future treatments will have to take this dimension into consideration, and MRD status will possibly be able to help better guide these progressions. 

Myeloma remains a deadly disease, although more and more patients live with a chronic status. Once patients have reached RRMM setting, become double and progressively quadri/penta refractory, MM enters the unmet medical need definition and death is inevitable, often in months. The constant progress brought to the early line developments will see an improvement of this condition, but certain drugs will always be offered to patients late in the disease course, and therefore, understanding the best MM management and sequencing of drugs will remain key to the treatment of myeloma.

## Figures and Tables

**Figure 1 cancers-12-02885-f001:**
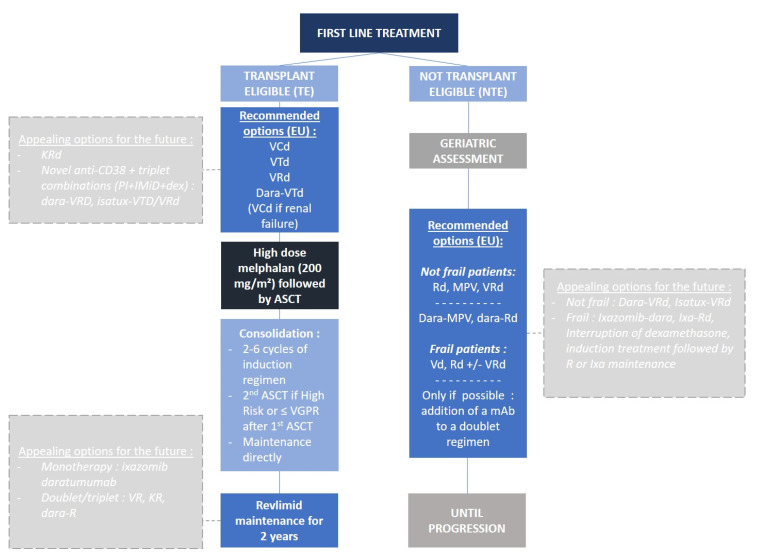
First-line treatment algorithm for transplant- and not transplant-eligible multiple myeloma patients.

**Figure 2 cancers-12-02885-f002:**
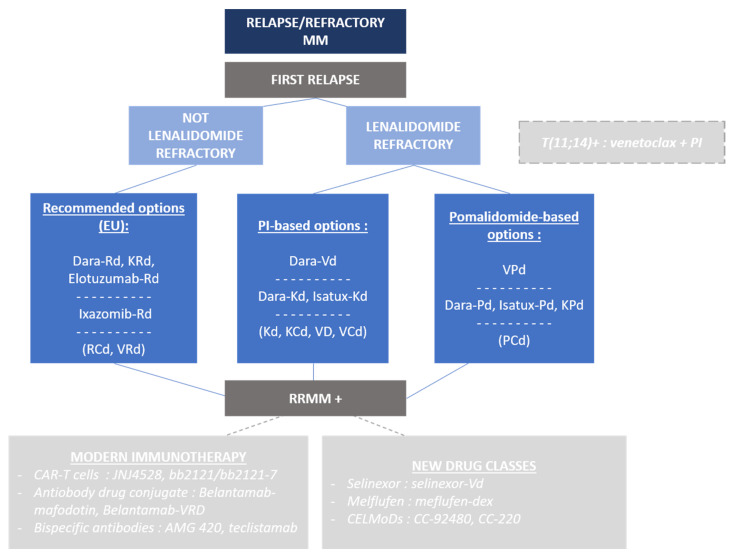
Relapse/refractory treatment algorithm for multiple myeloma patients.

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
