# Peer review of "Multiple Myeloma: An Overview of the Current and Novel Therapeutic Approaches in 2020"

_cancers, 2020, doi:10.3390/cancers12102885_

Round 1

Reviewer 1 Report

This review paper focused on the current status of novel treatment in patients with multiple myeloma. The authors summarized standard treatments in newly diagnosed and relapsed patients as well as promising agents under development. They recommended autologous stem cell transplantation as the frontline therapy in eligible patients. They also discussed the role of immunotherapy with novel antibodies and CAR-T cell therapy and future perspectives for cure.

For a better understanding, it is recommended to make a Table of novel drugs according to the categories and briefly comment on each feature.

Author Response

Dear Reviewers,

You will find below the responses to your comments.

We underlined the changes in the manuscript which will be attached to this letter.

Thank you for your comments about our work.

The authors.

Reviewer 1:

For a better understanding, it is recommended to make a Table of novel drugs according to the categories and briefly comment on each feature.

Thank you very much for your comments about our manuscript. Given the number of drugs, families and combination, a table would by nature be non-exhaustive. We chose to provide 2 figures describing the therapeutics paths instead of tables in order to illustrate our point.

Reviewer 2 Report

great summary paper

in the section where you review the data for the CTN-0702 Stamina trial

there has been an update as ASCO2020 from Dr. Hari

some of the findings were different than the original 38-month follow up dataset

specifically the PFS improvement in the ITT population for HR MM with tandem ASCT

the upfront IRd data has also been recently reported and should be added to the ixazomib section

Author Response

Dear Reviewers,

You will find below the responses to your comments.

We underlined the changes in the manuscript which will be attached to this letter.

Thank you for your comments about our work.

The authors.

Reviewer 2:

In the section where you review the data for the CTN-0702 Stamina trial there has been an update as ASCO2020 from Dr. Hari some of the findings were different than the original 38-month follow up dataset

specifically, the PFS improvement in the ITT population for HR MM with tandem ASCT

Thank you for your comment; we added the updated analysis. Page 5.

“the upfront IRd data has also been recently reported and should be added to the ixazomib section”.

Thank you for your comment; we added the results from this study as requested. Page 7

Reviewer 3 Report

Overall the content presented here is reflective of the present state of MM therapy. This is a difficult field to summarize in the aloted space. I do have some specific comments:

  1. For induction section, this is quite comprehensive with the available data but the focus on depth of response may be misleading. It is important to acknowledge that despite the suggestion that PI-Imid-steroid regimen is superior there is no comparative data examining PFS or OS. Further, there is a bias toward more European based-approaches such as VTd which is not a standard around the world thus the language should be tempered.
  2. In the ASCT section, it is true that the weight of evidence remains in favour for ASCT the inclusion of the more contemporary trials with KRD should be tempered. The data is quite preliminary and should be acknowledged that currently PFS and OS remains similar across the arms.
  3. In the maintenance section it should be acknowledged that the current standard is to deliver the lenalidomide until progression to maximize both PFS and OS benefit.
  4. In NTE section, should acknowledge that CyBorD remains an acceptable standard and that VMP is not the only PI-Alkylator-dex combination.
  5. In NTE section, should also make a comment regarding the recent IRd data as well as the ENDURANCE trial.
  6. In the RRMM would be cautious about how len-based therapies in len-exposed patients are effective given very small cohorts within the trials described and the fact that most patients will be refractory to len once exposed given how it is standardly used.
  7. I am not sure about the reference to checkpoint inhibitors being a "Phoenix" is warranted as the all the data in MM with available agents is negative.
  8. The CART data is well summarized. WOuld be beneficial to have a comment that depth of response may be less predictive of response durability and thus further followup in phase II and phase III studies are crucial to establish the palce of these products in MM.

There are a number of grammatical issues  that would benefit from a formal review of the manuscript from this perspective. It would improve the clarity prior to publication. Lastly, given the ongoing challenges with sequencing therapy in MM it would be helpful to provide a diagram summarizing the potential therapeutic paths in relapse based on sensitivity to prior therapy (ex len refractory vs PI-refractory as noted in the manuscript).

Author Response

Dear Reviewers,

You will find below the responses to your comments.

We underlined the changes in the manuscript which will be attached to this letter.

Thank you for your comments about our work.

The authors

Reviewer 3:

For induction section, this is quite comprehensive with the available data but the focus on depth of response may be misleading. It is important to acknowledge that despite the suggestion that PI-Imid-steroid regimen is superior there is no comparative data examining PFS or OS. Further, there is a bias toward more European based-approaches such as VTd which is not a standard around the world thus the language should be tempered.

The most commonly used regimen remains VTd for induction (32%), but mostly in Europe[1]. VTd was found superior to VCd in terms of very good partial response (VGPR) rates (VTd 66.7% vs 56.2% VCd, p=0.05). However, no direct comparison between VTd and VRd has ever been reported. Though, an integrated analysis of 4 randomized controlled trials (GEM2005, GEM2012, IFM 2009, IFM 2013-04) showed an improvement in ≥VGPR rate with VRd versus VTd, and VRd also led to less peripheral neuropathy and treatment related adverse events (AE)[2]. Thus, depth of response appears higher with VRd but the impact on survival is not demonstrated over VTd.

We have edited the chapter as follow. Page 3

In the ASCT section, it is true that the weight of evidence remains in favor for ASCT, although it might be challenged with the more contemporary trials with KRD. The data is quite preliminary and it should be acknowledged that currently PFS and OS remains similar across the arms. The improved sustained MRD negative rate in the ASCT arm over KRd no ASCT arm would let us believe that ASCT might remain the standard of treatment despite this induction improvement.

In the maintenance section it should be acknowledged that the current standard is to deliver the lenalidomide until progression to maximize both PFS and OS benefit.

Thank you, we added this notion. Page 5.

Currently, lenalidomide is mainly given until progression to maximize both PFS and OS, even though there has never been a direct comparison between fixed and continuous R maintenance. The 2 portions of the IFM2209 (fixed maintenance duration) DFCI (maintenance duration until progression) trial will provide some answer to this question.

In NTE section, should acknowledge that CyBorD remains an acceptable standard and that VMP is not the only PI-Alkylator-dex combination.

We added a phrase about the CyBor(1.5H)D/V(1.3 H or biH)Cd combination. Page 6.

In NTE section, should also make a comment regarding the recent IRd data as well as the ENDURANCE trial.

Thank you for your comment; we added the results from both studies as requested. Page 7

The addition of carfilzomib to Rd was also tested for NDMM NTE patients in the phase 3 ENDURANCE trial (KRd vs VRd), but this study did not meet the primary endpoint and KRd failed to improve the PFS[24]. In the end, this study validated the VRd combination as a standard of care for these patients in EU countries as a suitable and efficient regimen for older patients.”

The phase 3 TOURMALINE-MM2 trial has investigated ixa-Rd vs Rd for NDMM NTE patients but the first results did not meet primary endpoint, and although the median PFS of Ixa-Rd was higher at 35.3 months vs 21.8 months for the Rd group (p=0.073), this difference was not statistically significant [Takeda press release, 2020].

In the RRMM would be cautious about how len-based therapies in len-exposed patients are effective given very small cohorts within the trials described and the fact that most patients will be refractory to len once exposed given how it is standardly used.

Thank you for your comment; we tempered this part. Page 8

I am not sure about the reference to checkpoint inhibitors being a "Phoenix" is warranted as the all the data in MM with available agents is negative.

Thank you for your remark; we changed the title of this part. Page 11

The CART data is well summarized. Would be beneficial to have a comment that depth of response may be less predictive of response durability and thus further follow-up in phase II and phase III studies are crucial to establish the place of these products in MM.

Thank you for your comment; we added this point in the CAR-T part. Page 10

There are a number of grammatical issues that would benefit from a formal review of the manuscript from this perspective. It would improve the clarity prior to publication.

Thank you for your remark; the manuscript was formally revised and we hope we have fixed most of the grammatical issues.

Lastly, given the ongoing challenges with sequencing therapy in MM it would be helpful to provide a diagram summarizing the potential therapeutic paths in relapse based on sensitivity to prior therapy (ex len refractory vs PI-refractory as noted in the manuscript).

Thank you very much for your comment. We provided 2 figures describing the therapeutics paths. Page 3 and 6.

Reviewer 4 Report

I compliment the autors on their exhaustive review on the complex world of existing and upcoming myeloma treatments. 

I would encourage the authors to add a paragraphe on risk adapted strategies in MM (e.g. the GMMG CONCEPT trial) and to discuss the problem of under/overtreatment from the risk based approach (do not undertreat standard risk pts. they have the biggest benefit!). This aspect should also be highlighted in the otherwise good discussion of the FORTE trial. 

The DSMM XVII trial should also be added to the discussion of quadruplet inductions under current scrutiny. 

As the world of MM new treatments is indeeed complex an instructive figure of MOAs and tables comparing the cited papers with respect to risk factors , proportion of LEN ref, tripple ref, penta ref pts etc. and outcomes would be highly appreciated. 

Author Response

Dear Reviewers,

You will find below the responses to your comments.

We underlined the changes in the manuscript which will be attached to this letter.

Thank you for your comments about our work.

The authors.

Reviewer 4:

I would encourage the authors to add a paragraph on risk adapted strategies in MM (e.g. the GMMG CONCEPT trial) and to discuss the problem of under/overtreatment from the risk based approach (do not undertreat standard risk pts. they have the biggest benefit!). This aspect should also be highlighted in the otherwise good discussion of the FORTE trial. 

Thank you for your insightful comment, we added an entire paragraph about that topic. You will find it page 6 and down below:

Another important issue for the future of MM therapy is surely the risk-adapted strategy. High risk (HR) patients and standard risk (SR) patients should probably not be treated the same way. For the SR subgroup the tendency would be to treat less intensively and/or less longer. Studies are willing to demonstrate that ASCT could surely be questioned in patients who would reach an MRD negative status at the end of induction. Yet, the risk here is clearly to undertreat SR patients. Similarly, in this subset the need for a quadruplet induction regimen is also questionable as well as the type and duration of post-transplant consolidation and maintenance treatments. Concerning HR patients, the issue is completely different given the difficulty for most of HR patient to achieve an MRD negative status. Moreover, it also appeared that the treatment of HR patients in the relapse setting is extremely disappointing as well. Therefore, multiple studies enrolling HR patients are trying an “all-in” approach. The GMMG-CONCEPT trial (NCT03104842) brought the concept of treating patient with a quadruplet treatment based on their HR status (defined by the presence of del17p or t(4;14) or t(14;16) or > 3 copies 1q21 and ISS 2 or 3 stage disease). Isatuximab-KRd was given for 6 cycles of induction (+2 if no ASCT), 4 post-ASCT consolidation and isa-KR maintenance. Combining nearly all the drugs available in MM could probably be an interesting option for HR patients who have a poor outcome. The risk-based approach in MM is only starting and it might need more time to define which treatment regimen is more suited for which subset of patients, standard risk or high-risk patients. Plus, in the future we might potentially identify other risk-based subgroups (t(11;14), PET-CT negative, etc.)”

The DSMM XVII trial should also be added to the discussion of quadruplet inductions under current scrutiny. 

Thank you for your remark; we added this reference as well as other clinical trials focusing on quadruplet regimens. Page 4

As the world of MM new treatments is indeed complex an instructive figure of MOAs and tables comparing the cited papers with respect to risk factors, proportion of LEN ref, triple ref, penta ref pts etc. and outcomes would be highly appreciated. 

Thank you very much for your comment. We provided 2 figures describing the therapeutics paths. Page 3 and 6.